# Spatial Transformation Accelerator with Parallel Data Access Scheme for Sample Reconstruction

Rihards Novickis *[ID], Edgars Lielāmurs [ID], Daniels Jānis Justs [ID], Andrejs Cvetkovs [ID] and Kaspars Ozols [ID]

Institute of Electronics and Computer Science, 14 Dzerbenes St., LV-1006 Riga, Latvia;
edgars.lielamurs@edi.lv (E.L.); daniels.justs@edi.lv (D.J.J.); andrejs.cvetkovs@edi.lv (A.C.);
kaspars.ozols@edi.lv (K.O.)
* Correspondence: rihards.novickis@edi.lv

**Abstract:** Spatial image transformation is a commonly used component in many image processing pipelines. It enables the correction of optical distortions, image registration onto a common reference plane, electronic image stabilisation, digital zoom, video mosaicking, etc. With the growing tendency to embed image processing in low-power devices, attaining an efficient transformation solution becomes increasingly decisive. Furthermore, interpolation is the key operation in achieving the high quality of the transformed data from the original data. Fortunately, different implementations have already seen several efficiency improvements in recent years. However, interpolation relies on sampling a set of neighbouring points from memory, which has yet to be addressed efficiently for smaller computational platforms with limited memory resources. In this work, we derive a generic mathematical model and circuit design principles for the spatial transformation accelerator design for N-dimensional data. Furthermore, we present an efficient simultaneous access scheme for high-quality signal reconstruction. Finally, the introduced ideas are verified in field programmable gate arrays using one-dimensional and two-dimensional data transformation use cases. The presented solution is able to transform images with sizes ranging from $256 \times 256$ to $8192 \times 8192$ and achieves a transfer rate of 275 frames per second with $512 \times 512$ images.

**Keywords:** spatial transformation; digital circuit design; image processing; interpolation; resampling; acceleration; FPGA

## 1. Introduction

The inefficiencies in general-purpose processing combined with the end of Dennard scaling and Moore's law highlight the difficulties in further sustaining performance improvement rates [1]. Alongside improvements in compiler technology, domain-specific languages, caching hierarchy utilisation, etc., an alternative, more hardware-centric approach is to design tailored domain-specific architectures and accelerators.

For example, frequent limitations of the Internet of Things (IoT) technology [2] are size, weight and power. Such applications like security camera systems, autonomous vehicles and drones regularly depend on on-board processing due to latency, security and power consumption requirements. In response, the development of smart image sensor technology has emerged, which must support numerous combinations of image sensor specifications.

With the increasing chip heterogeneity, algorithm acceleration becomes a viable solution for improving the performance/watt ratio and can be exceptionally influential for frequently employed algorithms and their pipelined combination. This work examines the implementation prospects of such an algorithm: generic digital circuit design for spatial transformation with variable data dimensionality. Notably, the scientific literature offers a range of approaches concerning data resampling, distortion correction and interpolation.

## 2. Related Work

Different digital circuit-based resampling and interpolation methods have been explored [3–6]. For example, Aho et al. [3] demonstrated an accelerator design for image scaling, where pixels are resampled at a different resolution, and interpolation is necessary to restore the missing information. They present a detailed look into a parallel memory unit, separate address computation and data permutation blocks. The read and write addresses of each data element are computed with predetermined address functions. The permutation block organises data based on the address of each element. The work achieves a four-point window for bilinear interpolation, but increasing interpolation window size has not been demonstrated.

Zemčik et al. [4] introduces an efficient resampling algorithm, which is based on separable Finite Impulse Response (FIR) filtering and bi-linear interpolation. The solution applies to geometry distortion correction, where distortion is described through a rectangular mesh. The proposed solution separates vertical and horizontal resampling in independent pipelined modules with separate buffering schemes. The solution is demonstrated to be efficient. Nonetheless, it is only suitable for small geometrical distortions where the displacement is in the order of a few pixels.

A thought-out interpolation architecture based on a bi-cubic convolution interpolation algorithm with external memory is proposed by Mahale et al. [5]. Their solution utilises an elaborate buffering scheme and performs at 59 frames per second (using Xilinx Virtex-6). The authors address the under-utilisation of the computing resources used in interpolation and explore resource efficiency at the expense of reduced throughput. The authors manage to utilise external memory; nonetheless, the solution is only suitable for image down-scaling and up-scaling.

More recent bicubic interpolation architecture [7] manages to lower resources and improve the maximum frequency by replacing multiplications with summations and shift operators. A frequency of 289 MHz was achieved on a Xilinx Artix-7 Field Programmable Gate Array (FPGA) device with a fairly good interpolation quality compared to the software-based method. Boukhtache et al. [8] explored different alternative bicubic interpolation algorithms that reduce resource consumption while maintaining approximation accuracy. Multiplier count is reduced by more than 65% compared to standard bicubic implementations by combining cubic and linear interpolation algorithms. Similarly, only image scaling is possible in solutions proposed by Khaledyan et al. [7] and Boukhtache et al. [8] due to the utilisation of a simple sliding window-based buffer.

Chiew, Lin and Soon [9] introduced another alternative interpolation method called Negative Squared Distance (NSD) intended for embedded real-time reconstruction. Contrary to conventional methods, NSD is targeted towards FPGA-based implementation and possesses the characteristic of being advantageously described with 1D lookup tables (LUTs). Although advanced memory sampling for image scaling and rotation operations has been demonstrated to evaluate the quality of the proposed interpolation method, unfortunately, this work focuses on reconstruction and does not explain image sampling. In contrast, we focus on describing the memory access scheme and utilise standard interpolation methods.

D'Arco et al. [6] proposed a real-time resampling algorithm and circuit between an ADC and acquisition memory. The final solution utilises linear interpolation and is synthesised for FPGAs and Application Specific Integrated Circuits (ASICs). While the design supports arbitrary sampling rates, it does not account for the potential application of non-uniform resampling.

Also, multiple teams [10–12] proposed solutions that specifically address lens distortion correction. For instance, Ref. [10] have optimised fisheye distortion correction implementation for hardware by leveraging CORDIC [13] algorithms. Nonetheless, the actual distortion removal is managed by a Microblaze soft processor.

A relevant system has been proposed by Guo et al. [11], which handles real-time image distortion correction with a bilinear interpolation algorithm and a custom edge

enhancement scheme. Sensibly, the algorithm utilises a two-and-two-block Random Access Memory (RAM) buffering scheme to ensure parallel access. Their system achieves a real-time distortion correction at $1400 \times 1050$@60 Hz. Here, we adopt a similar approach while extending it to even higher dimensions and the number of memories.

Junger et al. [12] proposed a spatial transformation solution for radial lens distortion correction and image rectification of stereo-images. Their solution utilises a parallel access buffering scheme enabling bilinear interpolation, thus improving the quality of the output pixels. The transformation is precomputed and provided as an inverse transformation map, i.e., coordinate pairs of the input image. The transformation map is streamed from memory, and authors compress it to save the communication bandwidth. An alternative approach, showcased in this article, is to compute transformation on the go.

Our proposed work is a generic solution for carrying out a variety of data spatial transformations in digital logic. The proposed accelerator enables real-time computing, omitting the need for a communication bandwidth saving mechanisms as in [12] and enabling the fully digital implementation of such use cases as real-time image rectification and registration, digital zoom, lens distortion correction and compression [14]. Additionally, in this work, we formalise the logic behind parallel data access, which enables 16-point bicubic interpolation and improves over other related work: [11,12]. Furthermore, the derived model has the potential of also utilising parallel data access schemes for volumetric data.

### 3. Conceptual Design of the Spatial Transformation Core

*3.1. Functional Architecture of the Spatial Transformation Core*

Figure 1 illustrates the overall architecture of the spatial transformation accelerator. It consists of the following logic blocks: Input and Output Coordinate Counters, Dual-Port Memory Matrix, Write and Read Masters, Output Buffering Logic, Control Logic and external Inverse Transformation Computing. The accelerator accepts and provides data according to the Advanced Extensible Interface (AXI) Stream protocol. The spatial transformation accelerator's working principle relies on the sequential estimation of the consecutive output sample's location in the input image and reconstructing it using buffered input samples.

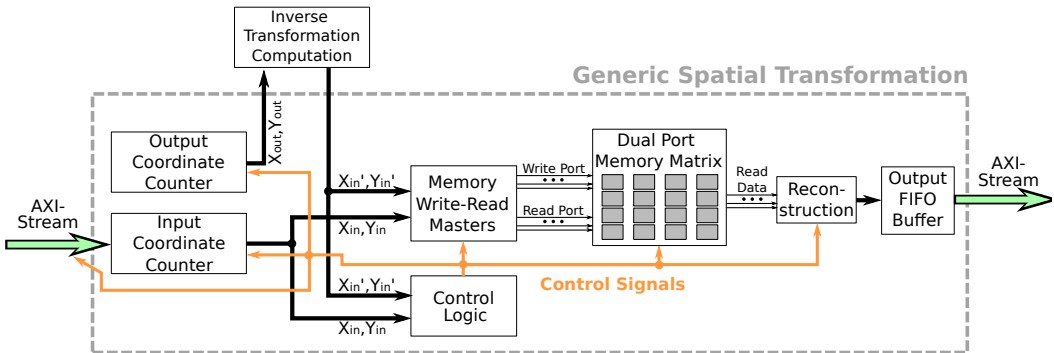

**Figure 1.** Functional architecture of the image spatial transformation accelerator.

In general, any image data transformation can be expressed as:

$$(x_{out}, y_{out}) = f(x_{in}, y_{in}), \tag{1}$$

where $x_{in}, y_{in}$ and $x_{out}, y_{out}$ denote the input/output image coordinates, and $f$ is some arbitrary function, often expressed as a matrix operator for linear transformations. Our proposed transformation solution necessitates the opposite: estimating the inverse transformation and essentially retrieving the corresponding input coordinate pairs for the consecutive output coordinates, i.e.,

$$(x_{in}, y_{in}) = f^{-1}(x_{out}, y_{out}). \tag{2}$$

Input coordinate retrieval is achieved by the Output Coordinate Counter and Inverse Transformation Computation blocks. Furthermore, the Input Coordinate Counter provides image coordinates for the input data stream; consequently, the Control Logic has access to the required and available data samples. This structure enables the simultaneous writing of input data to (buffer) memories and calculating the appropriate read addresses for the output data.

The final value for every output pixel can be reconstructed using adjacent pixels in the input image. Dual Port Memory Matrix and Memory Write-Read Masters ensure such functionality in a pipelined manner following a refined algorithm described in subsequent chapters. Notably, the Memory Read Master arranges a buffered pixel data for the Reconstruction, e.g., bi-linear or bi-cubic interpolation. The output First-In–First-Out (FIFO) buffer prevents the computing pipeline from stalling.

In some cases, consecutive coordinates from the transformed output image may result in coordinate "hopping", i.e., the coordinate pairs provided by the Inverse Transformation Computation block have relatively large distances in the input image. For example, in backward lens distortion coordinate mapping illustrated in Figure 2, each of the corrected output image coordinate pairs is used to calculate coordinates in the distorted input image. In severely distorted images, it equates to large $\Delta x$ and $\Delta y$ deviations. Hence, a sizable memory buffer is required, and the Memory Write-Read Masters should introduce a write-read distance threshold for potentially blocking the write data path.

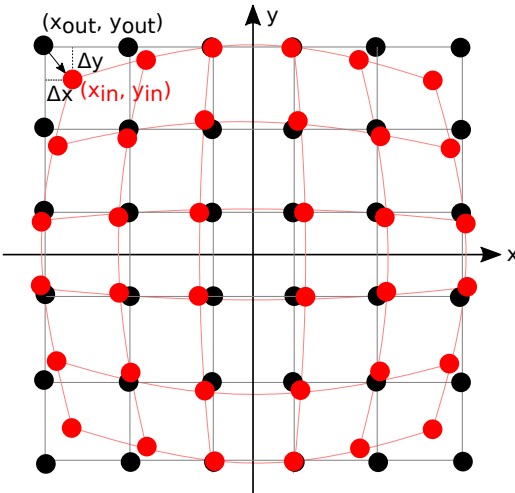

**Figure 2.** Coordinate mapping for Barrel distortion correction illustrating a potential coordinate "hopping".

The proposed accelerator concept partially resembles the solutions proposed in [3,9,12]. However, in this article, we derive an efficient parallel access scheme for sample buffering and retrieval. Furthermore, we generalise the model to N dimensions and demonstrate the concrete methodology behind deriving digital circuit designs. The presented approach enables pipelined spatial transformation and sample reconstruction utilising an additional number of input data points. Furthermore, the proposed solution is not constrained to a specific manipulation. The Inverse Transformation Computation represents a generic concept for computing coordinate mappings for any spatial transformation, e.g., in the case of image processing, lens distortion correction could be combined with linear transformation.

### 3.2. Memory Access Scheme

Figure 3 illustrates the overall approach for buffering and data access while performing the inverse transform. The write pointer continuously rolls over while, simultaneously, the read pointer is updated using the respective write and read requests. The implementation of such an approach is straightforward for a single memory use case where the output signal is reconstructed using a single sample, i.e., the nearest neighbour interpolation

method. Nonetheless, the requirement for signal reconstruction with multiple samples introduces additional complexity, yielding a compelling optimisation challenge.

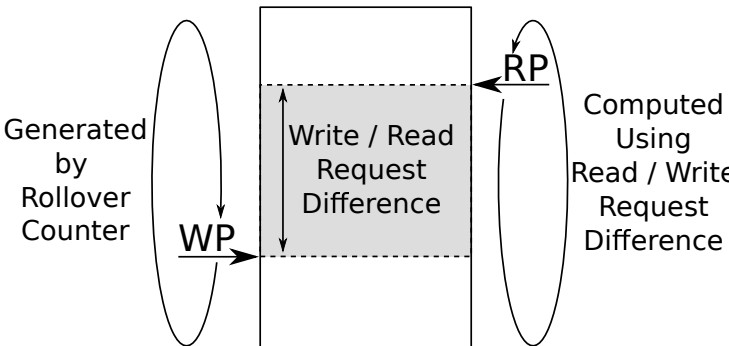

**Figure 3.** The overall concept of write and read pointer access to the memory.

One of the core requirements for achieving a fully pipelined design while enabling more advanced interpolation is a memory buffering scheme that permits interpolation logic with simultaneous access to adjacent pixels. The straightforward solution is based on the use of multiple dual-port memories, as shown in Figure 4. While this approach is convenient, it requires a considerable amount of memory, given that pixel data for interpolation are always adjacent. Figure 5 offers a more optimised solution, where the size of the memories is reduced by only storing the data for the respective odd and even columns and rows. While such an approach reduces the required memory size four times, it injects additional complexity for the write and read request multiplexing.

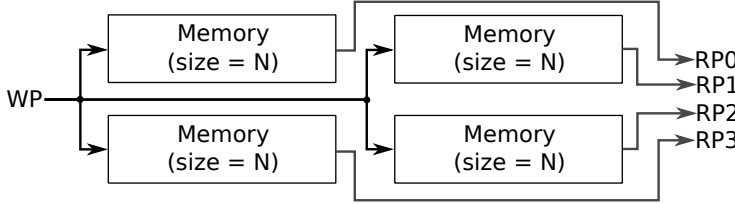

**Figure 4.** Straightforward data access scheme for 4-point reconstruction. Each memory Write Port (WP) processes an identical request while Read Ports (RPs) permit parallel access. Arrows denote the direction of the data flow.

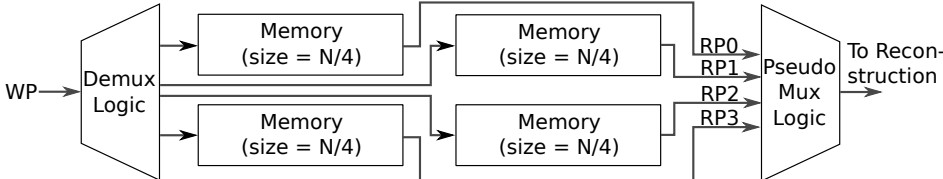

**Figure 5.** Optimised data access scheme for 4-point reconstruction, requiring 4× less memory. Each Write Port (WP) processes requests separately, while Read Ports (RPs) require additional logic for address computation and data reordering. Arrows denote the direction of the data flow.

Figure 6 demonstrates data sample storage when using the exemplary configuration illustrated in Figure 5. Note the different address combinations caused by the varying reconstruction areas. The modelling and generalisation of these address configurations are the main contributions of this article. The following sections disassemble the challenge, derive a mathematical model and disclose the general circuit generation principles.

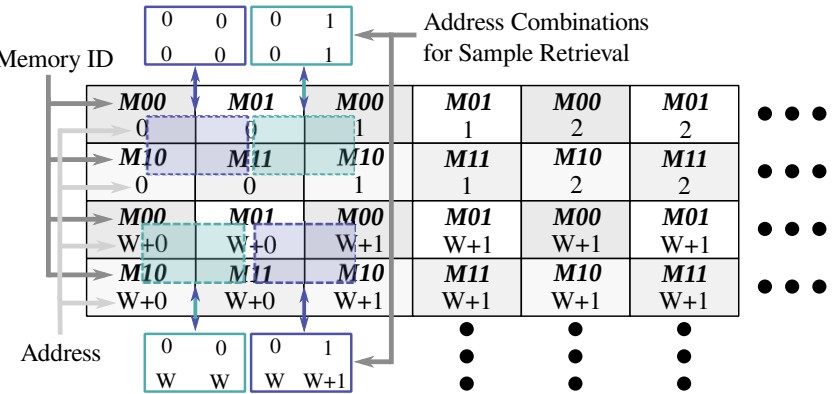

**Figure 6.** Storage of adjacent data samples across 4 memories in a 2 × 2 buffer configuration (dotted rectangles represent various reconstruction areas).

## 4. Derivation and Design of the Digital Circuit

### 4.1. Write Model and Circuit Generation

Typically, the corresponding memory address for the given coordinates adheres to the equation:

$$a = Y \times W + X, \tag{3}$$

where $a$ denotes the address, $W$ denotes the image width and $(Y, X)$ denote the image row and column. Nonetheless, the Generic Spatial Transformation circuit does not store the whole image in the On-Chip Random Access Memory (OCRAM). Utilising the consecutive quality of the write pointer, it is possible to substitute the write address generation logic with a simple rollover counter with the maximum value corresponding to the size of the memory buffer and the characteristics of the respective spatial transformation calculations. Still, we require row and column indices to compute the difference between the write and read pointers.

Considering that the memory matrix has $M$ rows and $N$ columns, the trailing $log_2(M)$ row signal bits and $log_2(N)$ column signal bits determine the demultiplexing functionality for the write pointer, illustrated by Figure 7. Naturally, for non-power-of-two reconstruction methods, the demultiplexing circuit becomes more complex.

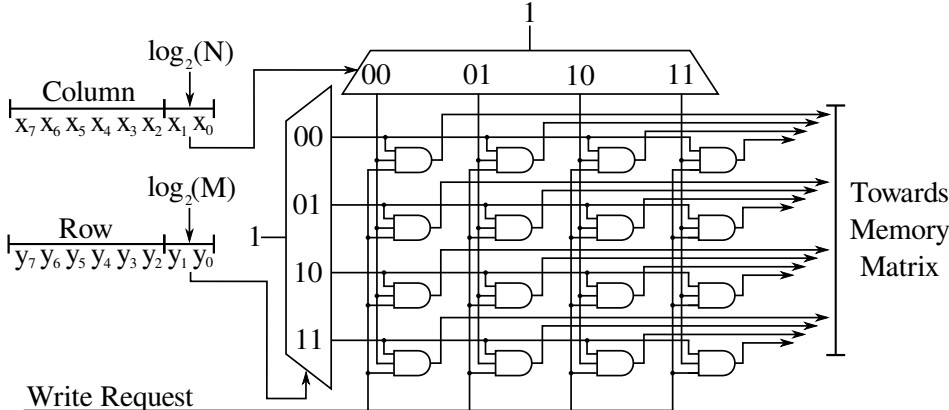

**Figure 7.** An example of write request demultiplexing logic when using a 4 × 4 memory matrix.

### 4.2. Read Model and Circuit Generation

Data retrieval is more challenging because of the requirement to generate varied addresses for each of the memories' read ports. Such a circuit design process can be formalised. First, let us assume a use-case for a one-dimensional data reconstruction with four samples, i.e., four memories. Figure 8 dissects such a one-dimensional request into $r_h$—high bits representing the core address for the sliced memories; $r_l$—low bits

utilised for individual memory address generation; and $r_f$—fixed-point part used later for reconstruction.

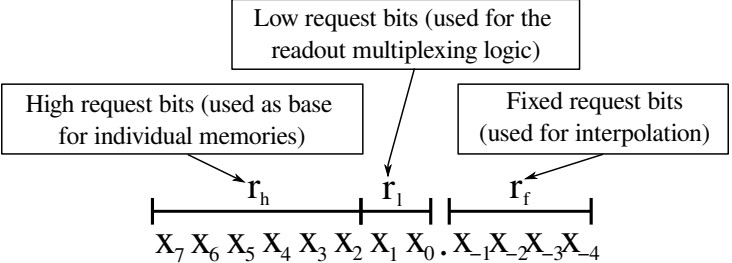

**Figure 8.** Dissection of a 12-bit one-dimensional reconstruction request for a 4-memory (4 sample) readout.

Figure 9 illustrates four variants of such a request for a single dimension where data samples reside across four memories. Here, the input sample index is denoted by $i$, and $a$ is the memory address in the corresponding memory. We consider the readout circuit to be an operator that takes the request and generates the address vector for the relevant data retrieval. Note that the address vector corresponds to the memories in the order $m0$–$m3$, and the retrieved data may require further reordering for a proper sample reconstruction, i.e., interpolation.

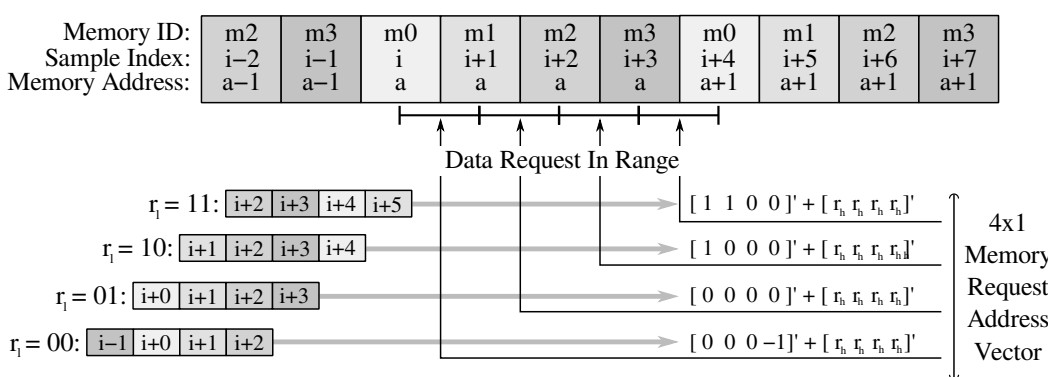

**Figure 9.** Data retrieval for reconstruction using four input data samples.

The demonstrated mechanism for data retrieval can be expressed in a matrix form with the following equation:

$$a = Sv_o + r_h, \tag{4}$$

where $a$ denotes the address vector, $\mathbf{S}$ denotes the shift matrix, $v_o$ denotes the constant offset vector and $r_h$ denotes the high bit portion of the request. In this particular 2D case, the offset vector would be:

$$v_o = \begin{bmatrix} 1 & 1 & 0 & 0 & 0 & 0 & -1 \end{bmatrix}' \tag{5}$$

and with $J$ being the reverse identity matrix, the shift matrix is:

$$\mathbf{S}_{4\times7} = \begin{cases} \begin{bmatrix} 0_{4\times3} & J_{4\times4} \end{bmatrix}, \text{if } r_l = 0 \\ \begin{bmatrix} 0_{4\times2} & J_{4\times4} & 0_{4\times1} \end{bmatrix}, \text{if } r_l = 1 \\ \begin{bmatrix} 0_{4\times1} & J_{4\times4} & 0_{4\times2} \end{bmatrix}, \text{if } r_l = 2 \\ \begin{bmatrix} J_{4\times4} & 0_{4\times3} \end{bmatrix}, \text{if } r_l = 3 \end{cases} \tag{6}$$

Furthermore, the offset vector can be constructed for any power-of-two number of memories $N$ as:

$$v_o = \begin{bmatrix} \underbrace{1\cdots1}_{N/2} & \underbrace{0\cdots0}_{N} & \underbrace{-1\cdots-1}_{N/2\text{-}1} \end{bmatrix} \tag{7}$$

The derived model can be extended to the second dimension by considering a specific signal reconstruction use case where reconstruction is explicitly executed across the vertical axis. Such constraint corresponds to stretching or contracting the image along the vertical axis. Figure 10 illustrates this particular case using four memories. The only difference compared to the horizontal reconstruction use-case is the offset added to the memory request address—which corresponds to the image width as follows:

$$C = \frac{image\ width}{log_2(number\ of\ memories)}. \tag{8}$$

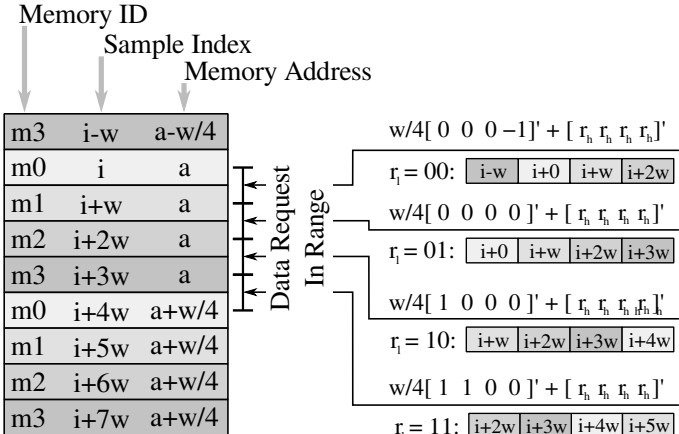

**Figure 10.** Data retrieval for reconstruction using four vertical input data samples.

The vertical retrieval mechanism can be expressed similarly to Equation (4) by introducing an offset constant $C$:

$$a = C\mathbf{S}v_o + r_h, \tag{9}$$

Further extending to any dimension, the address vector becomes:

$$a_n = C_n\mathbf{S_n}v_{o_n} + r_{h_n}, \tag{10}$$

where $C_1 = 1$ and $n$ is a dimension index.

Figure 11 illustrates the combination of vertical and horizontal retrieval mechanisms to generate the required addresses for all memories. Note that we have chosen to omit the offset constant from the offset vector in order to highlight the relationships for different output address matrices.

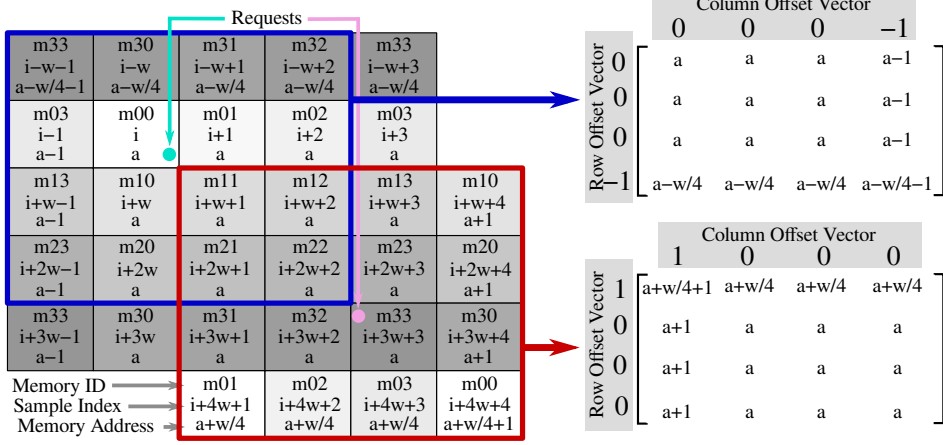

**Figure 11.** Address generation for a $4 \times 4$ memory matrix.

Finally, the derived mathematical model can be mapped into particular digital designs. For $N$ dimensions, memory addresses are expressible as an $N$-dimensional matrix $A \in E^N$, where each element is computed by:

$$A_{i_1 i_2 \cdots i_N} = \sum_{n=1}^{N} C_n S_n v_{0_{i_n}}. \tag{11}$$

Figure 12 encapsulates Equation (11) and illustrates the overall circuit design for $N$ dimensions. This incorporates the following major components:

- Offset Vector(s) —generates $S_n v_{0_{i_n}}$ by multiplexing slices of the constant offset vector (Equation (7)), as illustrated in Figure 13.
- Read Access Generation Circuitry—sets up all possible combinations for the memory matrix read addresses (3-value vector for the one-dimensional use case, $3 \times 3$ matrix for two dimensions, $3 \times 3 \times 3$ cube for three dimensions, etc.) and utilises previously generated offset vectors to multiplex addresses accordingly while consequently providing read addresses for all memories in the memory matrix. Figure 14 illustrates such circuitry for a two-dimensional use case. In the case of constant input dimensions, the circuitry becomes more efficient by preprocessing the fixed combinations.
- Memory Matrix—abstracts multiple memories corresponding to the number of data samples used for the reconstruction. In the current solution, each memory incorporates separate ports for writing and reading (i.e., T8 Static RAM (SRAM) cell); nonetheless, the continuous nature of the write requests could be exploited to utilise a single-port memory (i.e., T6 SRAM cell) and schedule simultaneous writes at the expense of introducing regular pipeline stalls.
- Demultiplexing Logic—rearranges read memory data for a consistent reconstruction of the output sample. Figure 15 illustrates such a circuit for a one-dimensional use case, and it resembles the opposite action of the multiplexing logic in the offset generation circuit (Figure 13). The design is reusable for every extra dimension.

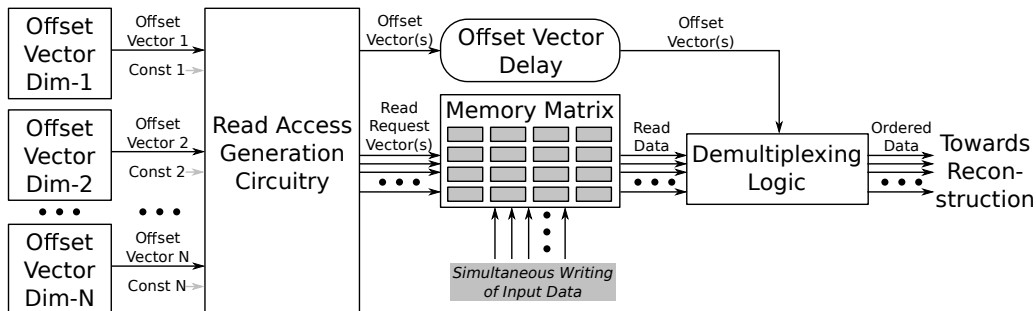

**Figure 12.** Read request generation for N dimensions.

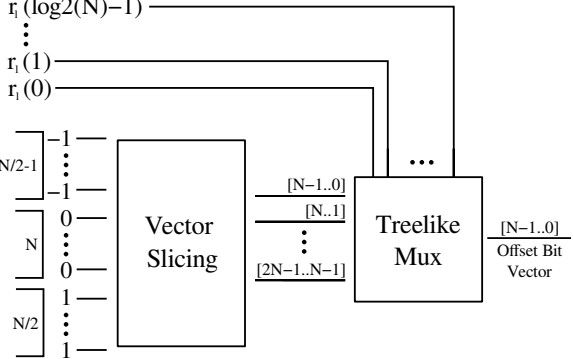

**Figure 13.** Circuit example utilising Equation (7) to construct $S_n v_{0_{i_n}}$ vector for a single dimension.

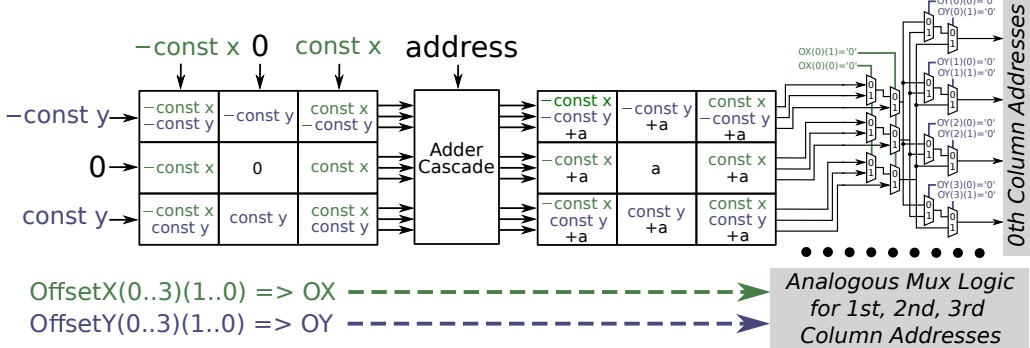

**Figure 14.** Conceptual design of read access circuitry for a two-dimensional use-case.

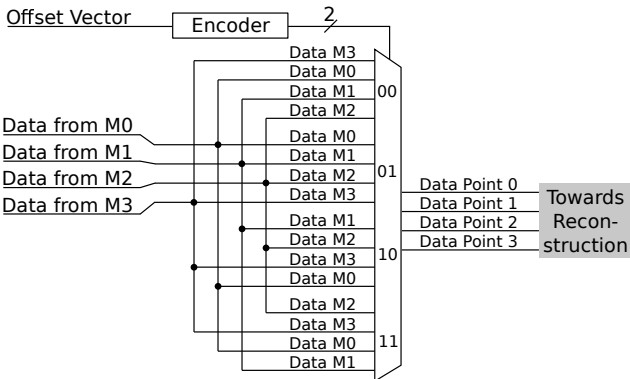

**Figure 15.** Conceptual design of the data rearrangement logic for one-dimensional use-case.

*4.3. Signal Reconstruction*

Since the memory access scheme provides configurable buffer matrix arrangements, different data reconstruction algorithms with varying neighbouring sample requisites are conceivable. The goal is to leverage the advantage of the optimised memory management strategy to implement cubic interpolation, one of the most accurate but more complex methods. Naturally, other interpolation methods are supported.

In cubic interpolation, a series of unique third-degree polynomials are fitted between each data point to obtain a continuous and smooth curve. To fit the curve and find the value at the coordinate $x$, we find appropriate coefficients in the following interpolation function:

$$f(x) = a_0 x^3 + a_1 x^2 + a_2 x + a_3, \tag{12}$$

where the four unknowns result from the four equations written using the four nearest neighbours of the point $x$.

The bicubic interpolation is a generalisation of the cubic interpolation for estimating data points on a rectangular grid. The value assigned to the reconstructed sample $(x, y)$ derives from the equation:

$$f(x, y) = \sum_{i=0}^{3} \sum_{j=0}^{3} a_{ij} x^i y^j, \tag{13}$$

where the sixteen coefficients correspond to the sixteen equations and sixteen nearest neighbours of the point $(x, y)$.

For hardware implementation, the cubic interpolation is calculated based on Equation (12) using fixed-point arithmetic operations. Four neighbouring sample values and the coordinate make up input data for the calculations and produce the reconstructed value. Figure 16 illustrates how five one-dimensional cubic interpolations on a rectangular grid result in bicubic interpolation. The first four interpolations reconstruct value in the horizontal direction. The final data point arises from the final vertical interpolation.

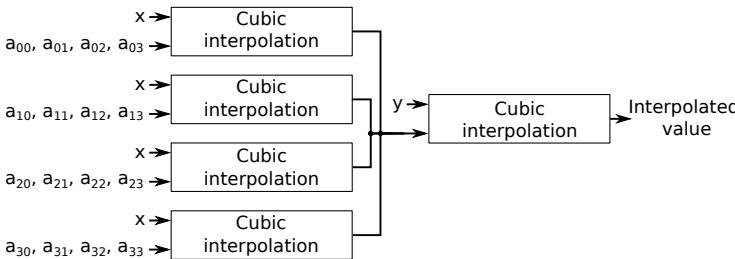

**Figure 16.** Bicubic interpolation composed of cubic interpolation subcomponents.

The proposed implementation is straightforward and sub-optimal concerning the minimisation of resource consumption because it is not the focus of this paper. For more efficient implementation variants, refer to [5,8].

## 5. Results and Evaluation

The spatial transformation controller concept has been implemented using Very High-Speed Integrated Circuit Hardware Description Language (VHDL) and evaluated by synthesising it with different buffer configurations. The hardware validation and concept verification process utilises Alpha Data *ADM-PCIE-8K5 Peripheral Component Interconnect Express* (PCIe) development platform containing Xilinx Kintex UltraScale KU115-2 FPGA. The provided transformation applications map the original and transformed coordinates to demonstrate the presented solution's performance. The component is accommodated with AXI Stream-compliant interfaces, while the PCIe *Direct Memory Access* (DMA) engine transfers data between the host and the device. Loopback tests enable performance (throughput) comparison against an empty system.

### 5.1. One-Dimensional Resampling

Resampling for one-dimensional data, such as audio recordings, was implemented using the cubic interpolation method to reconstruct the samples at the target coordinates. The generic spatial transformation accelerator, depicted in Figure 1, is configured to have four sample buffers in the Dual Port Memory Matrix to collect the neighbouring samples necessary for cubic interpolation. The coordinate processor utilises transformed sample coordinates to calculate the corresponding source sample location. In this example use-case, the processor can multiply coordinates with a coefficient or add an offset. Scaling the coordinates can resample any one-dimensional signal to a different sample rate, for example, to align sample rates (data registration) and prepare data for further processing.

Table 1 shows the implementation's resource utilisation on a Xilinx Kintex XCKU115-2-FLVA1517E device. Notably, the one-dimensional transformation use case necessitates only two block RAM units due to the small number of required buffered memory. Importantly, every Block RAM (BRAM) unit of the selected FPGA architecture is configurable as two independent 18 Kb RAMs [15], thus resulting in the utilisation of two (instead of four) BRAM units.

Table 1 also provides the maximum clock frequency and power consumption obtained from the Xilinx Vivado design suite. The data transfer rate is given as two measurements—the throughput of the bypass path and the throughput of the accelerator unit. Time was measured on the host machine by repeating a 10-megabyte (input) transfer 100 times and averaging the results. With a sample depth of 8 bits, the maximum capacity of the bypass path of the AXI stream channel was reached with a 125 MHz clock frequency. The transfer rates decreased notably to 833.83 Mbps when utilising the transformation accelerator.

**Table 1.** One-dimensional resampling hardware implementation results.

| DSP | Registers | LUT | BRAM | Frequency, MHz | Throughput (Bypass), Mbps | MSE |
|---|---|---|---|---|---|---|
| 5 (0.09%) | 812 (0.06%) | 634 (0.1%) | 2 (0.09%) | 236 | 833.83 (1041.03) | 0.1714 |

The accuracy of the interpolation was analysed by resampling a waveform and comparing the results with the exact solution. Figure 17 illustrates a rational conversion factor used to upsample a sine waveform from 11,025 to 44,100 samples/s. The in-between values are interpolated from the neighbouring samples, and the reconstructed waveform approaches the exact sinusoidal wave.

Generally, the latency is dependent on multiple factors, including the implementation of the Inverse Transformation Computing block, the chosen reconstruction method and the corresponding buffering scheme. Nonetheless, in the 1D use-case, the minimum achievable latency is 18 clock cycles, where two clock cycles are dedicated to constructing the reconstruction's sample coordinate, eight cycles are dedicated to storing samples in parallel buffers, two cycles are dedicated to reading and reordering samples and six cycles are dedicated to the actual reconstruction.

The results illustrate the concept's applicability for one-dimensional use cases. Nonetheless, the reconstruction precision depends on different aspects, such as fixed-point data-type precision and signal waveform.

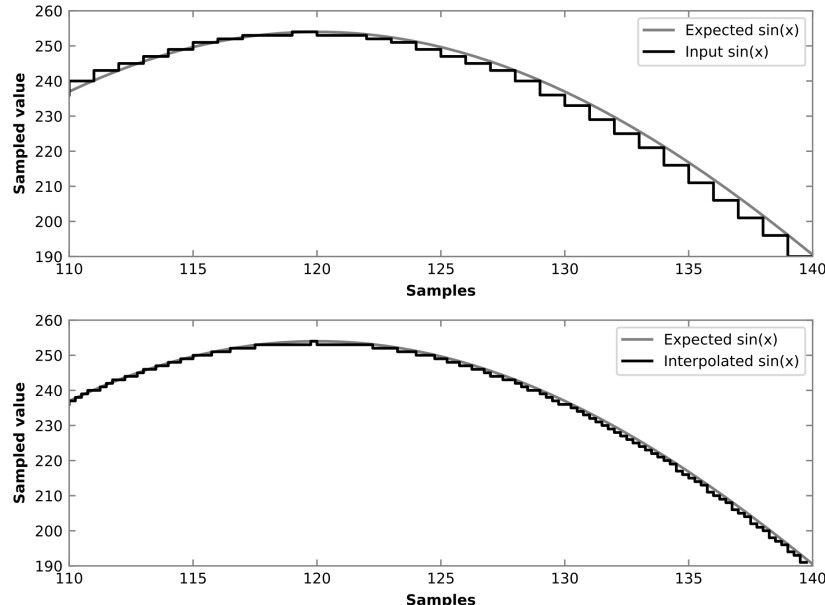

**Figure 17.** A linearly quantised test waveform represented with 8-bit integers is upsampled 4 times.

### 5.2. Two-Dimensional Resampling

For an image pixel resampling application, the system utilises 16 buffer memories, which ensures access to the input data samples for bi-cubic interpolation. A hardware matrix multiplier is employed to find the location of the source pixels by multiplying the target pixel coordinates with a transformation matrix. The coordinate processor can have numerous implementations; nonetheless, a matrix multiplier has the advantage of performing arbitrary transformations and provides a convenient mechanism for evaluating the proposed architecture.

Table 2 summarises resource consumption and provides a comparison with related work. The utilised resources with a single interpolation core and a matrix multiplier are 51 (0.92%) hardware multipliers, 5203 (0.78%) logic cells and 5304 (0.4%) registers. The overall usage of the embedded block RAM is variable and depends on the size of the dedicated buffer required for the specific transformation. For this purpose, the implementation utilises a 1 megabyte or 12.41% of available block RAM resources, which corresponds to the maximum buffering size of the accelerator. The core also is configurable for three interpolation cores, i.e., for RGB image reconstruction. In addition, we provide resource consumption without the matrix multiplier since the proposed architecture can achieve uncomplicated image transformations (e.g., scaling, translation) with simpler coordinate processors.

**Table 2.** Comparison of older bicubic interpolation designs with the proposed architecture and the proposed architecture with matrix multiplier (MM).

| Architecture | Method | DSP | Registers | LUT | BRAM | Frequency, MHz |
|---|---|---|---|---|---|---|
| Mahale et al. [5] (Gray) | Bicubic | 48 (5.0%) | 7843 (1.1%) | 7900 (2.2%) | 78 (12.3%) | 75 |
| Zhang et al. [16] (Gray) | Bicubic | 20 | 574 | 870 | - | 209 |
| Boukhtache et al. [8] (Gray) | 3C-2L | 11 | 2046 | 2470 | - | 212 |
| Proposed Gray MM | Bicubic | 51 (0.92%) | 5304 (0.4%) | 5203 (0.78%) | 256 (11.85%) | 173 |
| Proposed Gray | Bicubic | 23 (0.42%) | 4060 (0.31%) | 4050 (0.61%) | 256 (11.85%) | 185 |
| Proposed RGB MM | Bicubic | 95 (1.72%) | 10,841 (0.8%) | 10,184 (1.5%) | 768 (35.6%) | 172 |
| Proposed RGB | Bicubic | 63 (1.14%) | 9196 (0.69%) | 8385 (1.26%) | 768 (35.6%) | 204 |

Our proposed architecture without the matrix multiplier can be fairly compared with interpolation cores included in Table 2 considering that these works omit complex coordinate calculations. The implementation costs for the solution in [5] with intricate line buffers reveals that our proposed approach is more efficient. Other recent and optimised architectures [8,16] mentioned in a comprehensive study in [8] are less expensive but only include a shift buffer, which is only suitable for image scaling transformations. Notably, RGB implementations require roughly twice the number of LUTs and FFs compared to grayscale implementations. It can be explained by the scalability aspects of some of the components, in which resource consumption is determined by the resolution and pixel count rather than channel count, e.g., Coordinate Counters, Control Logic and Memory Write-Read Masters blocks.

Notably, while the memory-mapped architecture frequency is similar, the data show an unexpected anomaly. The RGB streaming architecture has a notably higher clock frequency than grayscale architecture, which is contrary to the expected. The inconsistency may be caused by the fitting algorithm, which employs some uncertainty.

Figure 18 depicts the transfer rates measured at several image resolutions. The estimation of the maximum transfer rates possible with the test environment utilised in this work follows the measurement of transfers while bypassing the accelerator (data loopback transfer). We select $2\times$ scaling and vertical shear transformations to expose the effects on the transformation performance while necessitating different amounts of buffer memory. Only a negligible delay arises during the scaling operation since the transformed coordinates increase consecutively. On the other hand, when a shear coefficient $\lambda$ for the $y$ dimension is present in the transformation matrix, the transformed coordinates are $y' = \lambda x + y$. As can be seen, a shear coefficient smaller than zero will cause the memory controller to sample each consecutive pixel with a decreasing row offset, necessitating a buffer, which can hold $|\lambda W|$ number of rows, where $W$ is the width of the image. Subsequently, the transfer rates, in this case, are lower. With the RGB arrangement, the transfer rates are slightly lower than with a single interpolation core but still reach 29 frames per second with a frame size of $2048 \times 2048$ pixels.

The reconstruction quality is evaluated by inspecting an image after resizing from VGA to XGA resolution, as shown in Figure 19. With a conversion factor of 1.6, only every fifth pixel is reused from the input data as it is, while others are interpolated. Stairstepping can be observed on the white blossom edges in the cropped original image in Figure 19c, whereas in the up-scaled image in Figure 19d, the effect is not noticeable. To visualise and compare the interpolation quality Figure 20 illustrates the absolute difference obtained by upscaling and then downscaling the VGA reference image by the same factor of 1.6. The mean square error (MSE) is used to evaluate the error compared to the reference image of the size N × M by finding the mean of the squared differences between each of the reference pixel values $p(i, j)$ and the generated pixels $p'(i, j)$:

$$\text{MSE} = \frac{1}{NM} \sum_{i=0}^{N-1} \sum_{j=0}^{M-1} (p(i, j) - p'(i, j))^2 \qquad (14)$$

We see that, in Figure 20, the proposed algorithm yields a 2.7 times higher MSE due to the reduced precision of the fixed point arithmetic.

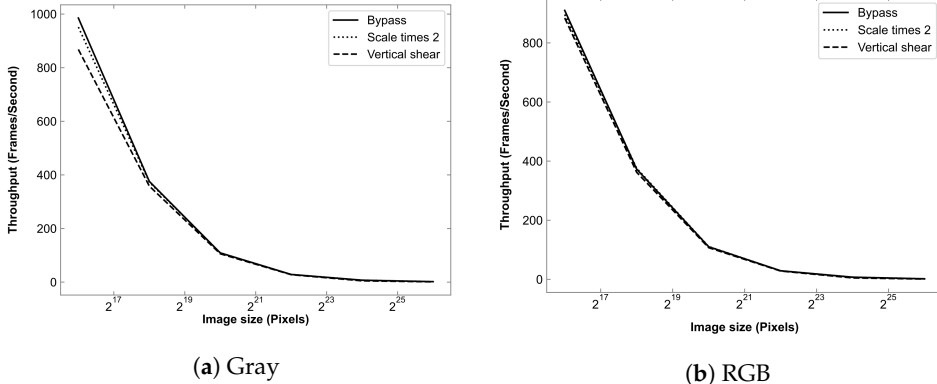

(**a**) Gray

(**b**) RGB

**Figure 18.** Rate of transformation at image sizes ranging from $256 \times 256$ to $8192 \times 8192$. (**a**) Single interpolation core. (**b**) Three interpolation cores.

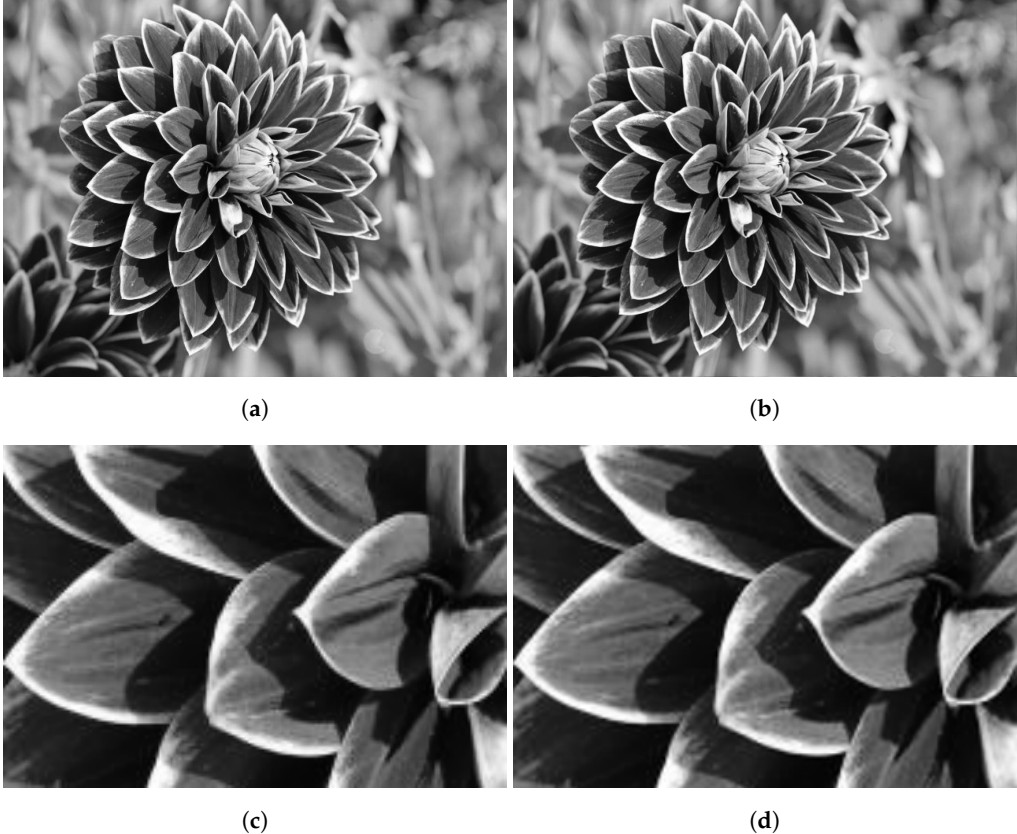

(**a**)                             (**b**)

(**c**)                             (**d**)

**Figure 19.** (**a**) Before scaling $640 \times 480$ image and (**b**) after 1.6 times up-scaling $1024 \times 768$ image. (**d**) Closer inspection of reconstruction after resizing and (**c**) initial for reference.

The results show that the interpolation error of the algorithm changes depending on the details in the test image. To fairly evaluate the quality of interpolation, we select five commonly used test images with varying levels of detail and perform different types of transformations. MSE and peak-signal-to-noise-ratio (PSNR) measurements relative to a software bicubic interpolation in Table 3 confirm that the proposed solution consistently provides low levels of errors.

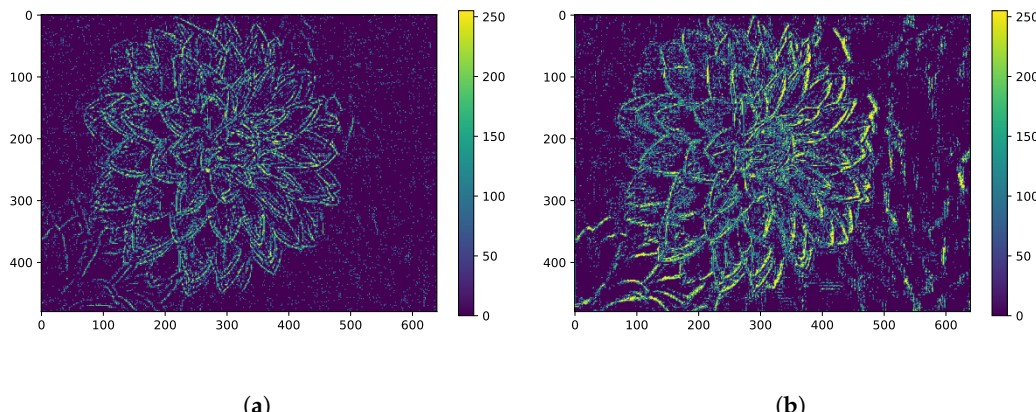

(**a**)                                                         (**b**)

**Figure 20.** Absolute error after upscaling and then downscaling a reference image by a factor of 1.6. (**a**) Reference image after software bicubic interpolation (MSE = 0.3924). (**b**) Image after proposed fixed-point interpolation (MSE = 1.0763).

Image rotation measurements in Table 3 and the result in Figure 21 show that the proposed architecture is suitable for transformation with deep read operations. In contrast to the line buffer approach, the parallel memory access scheme can store large fragments of the image or even an entire image. In the case of Figure 21, for a 45° rotation, at most half of the 512 × 512 image resides in buffers while maintaining a transfer rate of 275 frames per second.

The latency characterisation of a 2D use case is far more involved compared to the 1D scenario. Similarly, apart from the particular implementation of the Inverse Transformation Computing block, the reconstruction method and the corresponding buffering scheme, the latency is also influenced by the highest deviation in the transformation and nonlinearity aspects. For the simplest identity transformation use-case, the minimum achieved latency is 220 clock cycles, where 10 clock cycles are spent on computing the coordinate pair, 196 cycles—for storing samples in memory buffers, 2 cycles—for reading and reordering samples and 12 cycles—for the actual reconstruction.

**Table 3.** MSE and PSNR compared to software bicubic interpolation solution.

|  | Images | MSE | PSNR [dB] |
|---|---|---|---|
| Scale 1.25 | Cameraman | 1.7094 | 45.8024 |
|  | Lake | 4.1096 | 41.9928 |
|  | Lena | 2.1119 | 44.8842 |
|  | Peppers | 2.2535 | 44.6021 |
|  | Bridge | 7.4841 | 39.3894 |
| Rotate 30° | Cameraman | 2.4567 | 44.2273 |
|  | Lake | 4.2654 | 41.8312 |
|  | Lena | 2.5701 | 44.0313 |
|  | Peppers | 2.6491 | 43.8998 |
|  | Bridge | 7.6319 | 39.3045 |
| Rotate 45° | Cameraman | 1.6840 | 45.8675 |
|  | Lake | 2.9820 | 43.3857 |
|  | Lena | 1.9591 | 45.2103 |
|  | Peppers | 1.9436 | 45.2447 |
|  | Bridge | 5.6992 | 40.5727 |

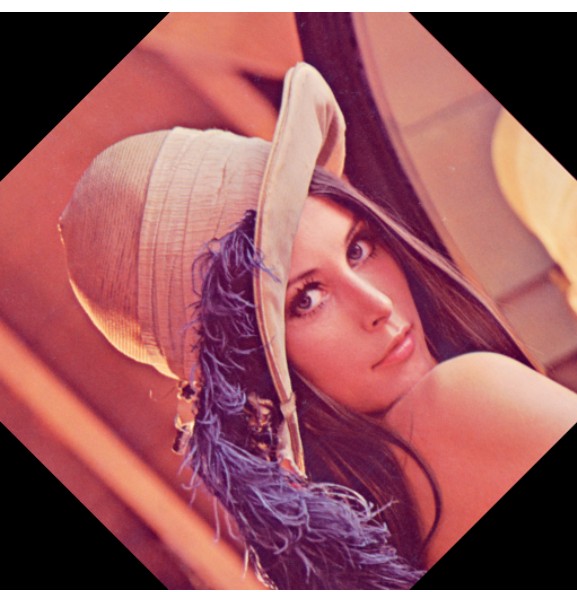

**Figure 21.** RGB image rotation demonstrates that transformations with deep buffering and three interpolation cores are attainable.

### 6. Conclusions

In this work, a parallel memory access approach for data resampling and reconstruction has been proposed. The concept consists of a mathematical model and digital implementation of the corresponding operators. The underlying address generation procedure manifests through a set of matrix operations and is applicable for an arbitrary number of data dimensions. Furthermore, the operator structure lends itself to parallel implementation mediums, e.g., programmable logic devices.

The scalability and performance were verified by applying the proposed concept for one- and two-dimensional data formats using cubic and bicubic interpolation methods. Hardware resource consumption, interpolation quality and transfer rates were evaluated using practical image scaling and rotation applications. Interpolation quality was estimated using MSE and PSNR measurements for comparisons with equivalent existing solutions. Results indicate that the interpolation achieves a pipelined performance and satisfies the reconstruction quality. Nonetheless, the overall resource utilisation is slightly higher when compared to related work due to more generic computational logic.

Compared to the less complex sample acquisition architectures, this technique provides deep read requests and justifies higher implementation costs by supporting resampling tasks with sparse address inquiries. This capability facilitates usage in different transformation computation instances, e.g., lens distortion correction and data registration. Most importantly, the proposed architecture ensures the efficient utilisation of the expensive on-chip (SRAM) memory resources.

Future advancement of the solution could incorporate a DDR-based memory controller(s) and comprehensive caching hierarchy. Such an approach could enable the real-time spatial transformation of enormous images and other data formats. Furthermore, the solution could be probed against currently unaccelerated higher dimensionality use cases.

**Author Contributions:** Conceptualisation, R.N.; Supervision, R.N.; Methodology, R.N.; Investigation, R.N. and E.L.; Writing, R.N. and E.L.; Review and editing, R.N., E.L. and K.O.; Original draft preparation, R.N. and E.L.; Implementation, E.L., D.J.J. and A.C.; Software, E.L.; Verification, E.L.; Measurements, E.L.; Visualisation, E.L.; Funding, K.O. All authors have read and agreed to the published version of the manuscript.

**Funding:** The results presented in this work are related to activities within the APPLAUSE project, which has received funding from the ECSEL Joint Undertaking (JU) under grant agreement No 826588. The JU receives support from the European Union's Horizon 2020 research and innovation programme and Belgium, Germany, Netherlands, Finland, Austria, France, Hungary, Latvia, Norway, Switzerland and Israel.

**Institutional Review Board Statement:** Not applicable.

**Informed Consent Statement:** Not applicable.

**Data Availability Statement:** Data are contained within the article.

**Conflicts of Interest:** The authors declare no conflict of interest. The funders had no role in the design of the study; in the collection, analyses, or interpretation of data; in the writing of the manuscript; or in the decision to publish the results.

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
