# Peer review of "Spatial Transformation Accelerator with Parallel Data Access Scheme for Sample Reconstruction"

_electronics, doi:10.3390/electronics13050922_

Round 1

Reviewer 1 Report

Comments and Suggestions for Authors

In this paper, the authors claim to present a hardware accelerator dedicated to the computation of spatial transformations. Although the idea of using multiple line buffers for computing spatial transformations is not new, the authors write that the modelling and generalization of the address configurations are the central contributions of this paper.
The paper is well written and well organized. The goal of the research presented in the paper is interesting but some points need to be enhanced.

Contrary to the claim, I don't clearly see the modelling and generalization in the current paper. Maybe an algorithm giving the principal characteristics of the on-chip buffering and read-write sub-system (figures 7-11) for a given transform and image size should help.
In order to avoid confusion with the word axis, I suggest replacing AXIS in figure 1 with AXI-Stream.
For a given maximum (DeltaX,DeltaY), a given image width W and a Number of dimensions N, the authors should give a formula for the size and number of memory buffers.
line 147: "For example, in image processing, the accelerator supports linear transformations and lens distortion correction as long as the Inverse Transformation Computation block supports it." I think the authors should list some commonly used transfomations or correction supported and unsupported (or at least the properties required in order to allow the inverse transformation computation).
line 268: "proposed implementation is straightforward and sub-optimal" sub-optimal with respect to which criteria?
Table 1 is a bit confusing: is the 236MHz the frequency of the FPGA design's clock? If yes how can the bypass and resampler throughput differ?
line 297: "by repeating a 10-megabyte (input) transfer 100 times" it would be usefull to also give the extrem values and at least the worst-case throughput.
Paragraph 4.1 I assume that the authors use equation (12), but the values and fixed-point encoding of the unknowns are not given.
Paragraph 4.2 is very ambiguous. Notably the sentence on line 324 "In addition, we provide resource consumption without the matrix multiplier since the proposed architecture can achieve uncomplicated image transformations (e.g., scaling, translation) with simpler coordinate processors". Does this mean that in figure 1 the "Inverse Transformation Computation" block is by-passed (Xin'=Xout and Yin'=Yout) or that a simpler coordintate processor is used? In the later case I don't understand why its resources usage is not taken into account for a fair comparison?
The results do not show the latency of the system, this is an important characteristics and need to be added. For example, if one would use the system for computing the output of a 5x5 image convolution kernel, I assume that the latency would be at least the time for reading 4 lines plus 5 pixels?
Table 2: the characteristics (gray or RGB) of the older bicubic interpolation designs should be added.
It is difficult to compare the works if the throughputs and latencies are not given.
It is surprising that "Proposed RGB" requires roughly twice the number of LUTs and FFs than the "Proposed Gray". One should expect roughly three times the number, do the authors have an explanation for this?
The discussion about the results (lines 317-332) does not really summarize the comparison of the results with other work. Does the presented architecture uses more resources but allows higher throughput for example (and at which cost? For example 2 times the resources with 3 times the throughput is not the same as 2 times the resources with 1.5 times the throughput).
It is also surprising that "Proposed RGB" achives a higher frequency than the "Proposed Gray", a similar frequency seems more intuitive, could the authors explain this?

line 342: "necessitating a buffer, which can hold λW number of rows" should be "-λW" or "|λW|" since λ is negative.

Table3: are the PSNR given as absolute values or in dB?

There are also some typos that need to be corrected:
line 164: (odd / pair) should probably be (odd / even)?
Fig 5: Pseido Mux Logic should be Pseudo Mux Logic
line 170: "the centre contributions" should be "the main (or central) contributions"

Author Response

Contrary to the claim, I don't clearly see the modelling and generalization in the current paper. Maybe an algorithm giving the principal characteristics of the on-chip buffering and read-write sub-system (figures 7-11) for a given transform and image size should help.
Thank you! The comment highlights a possibility for us to make our work more understandable. Figures 7-11 and the corresponding description illustrate the general concepts using a 2D processing approach, which is the simplest employable example for illustrating the emerging complexity of the data buffering schemes. The actual generalization starts with Figure 12, which extends the approach and circuit generation for N dimensions. We have added clarifications throughout the article (also your other comments), which hopefully clarifies this aspect in the article.

In order to avoid confusion with the word axis, I suggest replacing AXIS in figure 1 with AXI-Stream.
Thank you, we have updated the description accordingly.

For a given maximum (DeltaX,DeltaY), a given image width W and a Number of dimensions N, the authors should give a formula for the size and number of memory buffers.
Thank you! The number of dimensions N (memory count) is determined by the chosen interpolation method. While maximum deviation (DeltaX, DeltaY) indeed effect the size of the memories, which in current design corresponds to (DeltaX + ImageWidth*(DeltaY+4)). "+4" can be reduced by slightly increasing the circuit complexity. We have updated the article accordingly.

For example, in image processing, the accelerator supports linear transformations and lens distortion correction as long as the Inverse Transformation Computation block supports it." I think the authors should list some commonly used transfomations or correction supported and unsupported (or at least the properties required in order to allow the inverse transformation computation).
The Inverse Transformation Computation represents a generic concept for computing coordinate pairs with any imaginable spatial transformation, e.g. in our group a common application is to combine lens distortion correction with homography. For the sake of clarity, we clarified this aspect in the article.

line 268: "proposed implementation is straightforward and sub-optimal" sub-optimal with respect to which criteria?
The interpolation logic is sub-optimal concerning the minimization of resource consumption. This note has also been added in the article.

Table 1 is a bit confusing: is the 236MHz the frequency of the FPGA design's clock? If yes how can the bypass and resampler throughput differ?
Indeed, 236MHz is the maximum FPGA clock frequency for the digital design. Nonetheless, the shown throughput was measured using an actual Xilinx Kintex XCKU115-2-FLVA1517E FPGA device, and, naturally, the AXI-Stream channel saturates. The throughput is provided to assess the expected performance for an actual physical implementation.

Paragraph 4.1 I assume that the authors use equation (12), but the values and fixed-point encoding of the unknowns are not given.
No fixed-point encoding has been used because, as it is described in 4.1 paragraph, the sample representation itself is upscaled from 8-bits to 12-bits while sample rate has been increased by a fixed 4X ratio.

Paragraph 4.2 is very ambiguous. Notably the sentence on line 324 "In addition, we provide resource consumption without the matrix multiplier since the proposed architecture can achieve uncomplicated image transformations (e.g., scaling, translation) with simpler coordinate processors". Does this mean that in figure 1 the "Inverse Transformation Computation" block is by-passed (Xin'=Xout and Yin'=Yout) or that a simpler coordintate processor is used? In the later case I don't understand why its resources usage is not taken into account for a fair comparison?
Indeed, the resource consumption in Table 2 is provided without the "Inverse Transformation Computation" block (as if it is bypassed). In fact, it is omitted precisely to enable fair comparison with other solutions because the "Inverse Transformation Computation" block can be designed in numerous ways depending on the use case. For example, as it is described in the 4.2 paragraph, a simple scaling operation can be implemented without a matrix multiplier.

The results do not show the latency of the system, this is an important characteristics and need to be added. For example, if one would use the system for computing the output of a 5x5 image convolution kernel, I assume that the latency would be at least the time for reading 4 lines plus 5 pixels?
In general, only the minimum latency (when "Inverse Transformation Computation" block is bypassed) can be provided as in the case of non-linear transformations the latecany will vary greatly (Future 2 illustrates this aspect).

Table 2: the characteristics (gray or RGB) of the older bicubic interpolation designs should be added.
Thank you, we have updated the table accordingly.

It is difficult to compare the works if the throughputs and latencies are not given.
You are correct. The solution is fully-pipelined. Often this would result in bandwidth being strictly determined by the on-chip communication capabilities and maximum clock frequency. Nonetheless, the actual bandwidth and latency is very dependent on the specific use-case. For 1D resampling use-case the bandwidth has been shown to be 833.83 Mbps, while for 512x512 image rotation by 45 degrees 550 Mbps (275 fps). As for the latency, we have added the minimum achievable latency in terms of the clock cycles. Further it is determined by the "Inverse Transformation Block" and buffering requirements.

It is surprising that "Proposed RGB" requires roughly twice the number of LUTs and FFs than the "Proposed Gray". One should expect roughly three times the number, do the authors have an explanation for this?
Thank you for valuable comment. This difference is explained by the fact that resource count for some of the components is not influenced by the channel count, but rather by the resolution and buffer size in term of pixels, e.g. "Coordinate Counters", "Control Logic", "Memory Write-Read Masters" blocks. We have also added this clarification to the 4.2 section of the paper.

The discussion about the results (lines 317-332) does not really summarize the comparison of the results with other work. Does the presented architecture uses more resources but allows higher throughput for example (and at which cost? For example 2 times the resources with 3 times the throughput is not the same as 2 times the resources with 1.5 times the throughput).
As mention previously, such evaluation is less straightforward than one may expect due to the variety of different use cases and employed approaches. Nonetheless, our proposed we have added improvements throughout the article to highlight this aspect and also added latency / bandwidth characterization.

It is also surprising that "Proposed RGB" achives a higher frequency than the "Proposed Gray", a similar frequency seems more intuitive, could the authors explain this?
This is a good obeservation. Notably, this is true for Streaming (not Memory Mapped) configuration. Synthesized circuit's fitting algorithm has some uncertainty in it, which may have caused this strange result. We have also added this possible cause to the 4.2 section's discussion.

line 342: "necessitating a buffer, which can hold λW number of rows" should be "-λW" or "|λW|" since λ is negative.
Thank you, we have updated the description accordingly.

Table3: are the PSNR given as absolute values or in dB?
Thank you, the values are in dB, the table has been updated accordingly

line 164: (odd / pair) should probably be (odd / even)?
Thank you, we have updated the description accordingly.

Fig 5: Pseido Mux Logic should be Pseudo Mux Logic
Thank you, we have updated the diagram accordingly.

line 170: "the centre contributions" should be "the main (or central) contributions"
Thank you, we have updated the description accordingly.

Reviewer 2 Report

Comments and Suggestions for Authors

The paper suggests a useful concurrent access system for high-quality signal reconstruction designed for spatial image transformation.

The paper is nice and I enjoyed reading it; however, I have several concerns:

1. The Introduction and Related Work section is very long and can be divided into two parts, one that describes the contribution of this work and one that describes what has been done in the past in the field.

2. The abbreviation FPGA and other abbreviations are not defined. The paper needs a table of acronyms to facilitate the search for their definition.

3. In Figures 4 and 5, the authors need to specify what information goes through each arrow drawn in the figure.

4. In equation 3, what is a?

5. In Figure 6, there are several types of addresses and it is not clear what the difference between the addresses is, who produces the addresses and according to which algorithm. There is almost no explanation for this figure and it makes it difficult to understand.

6. What is the difference between W and W+0?

7. In a new paper Rakhmanov A., (2023), "Compression of GNSS Data with the Aim of Speeding up Communication to Autonomous Vehicles", Remote Sensing, MDPI, Vol. 15(8), paper no. 2165, available online at: https://www.mdpi.com/2072-4292/15/8/2165 , the author suggests in Figure 1 to interpolate the skipped data by an average from previous and subsequent information. I would encourage the authors to cite this paper in the related work section and explain what the advantages and disadvantages of this method are (compared to their methods).

8. The equations and the results seem to be detached. Please explain how you have designed the experiments based on the theoretical background.

9. The format of references should be consistent.

Author Response

1. The Introduction and Related Work section is very long and can be divided into two parts, one that describes the contribution of this work and one that describes what has been done in the past in the field.
Thank you! We have updated the article accordingly.

2. The abbreviation FPGA and other abbreviations are not defined. The paper needs a table of acronyms to facilitate the search for their definition.
Thank you for the suggestion, we have added the missing abbreviations, but have not added a separate table, as to our understanding, that would contradict journal formatting (for such a medium-length article).

3. In Figures 4 and 5, the authors need to specify what information goes through each arrow drawn in the figure.
Write Port (WP) and Read Port (RP) denote a memory-mapped connection that corresponds to a low-level interface/protocol, which is typically used for memory, it usually incorporates signaling at least for address, write/read data and write request. The direction of arrows represent the direction of the data flow. This aspect has been clarified accordingly.

4. In equation 3, what is a?
'a' stands for the memory address where (pixel) given its 'x' and 'y' coordinates resides.

5. In Figure 6, there are several types of addresses and it is not clear what the difference between the addresses is, who produces the addresses and according to which algorithm. There is almost no explanation for this figure and it makes it difficult to understand.
Thank you for pointing out potential improvement for the article. We have improved Figure 6 accordingly. There is a single memory type, the figure represents layout of the image in 4 memories optimized for bi-linear (4-point) interpolation. Given a need to reconstruct a the value in between adjacent pixels, we simultaneously retrieve adjacent pixels from multiple memories. Nonetheless, the figure illustrates that calculating these addresses is not straightforward.

6. What is the difference between W and W+0?
There is no difference, we added "+0" following a personal preference for a more symmetrical representation.

7. In a new paper Rakhmanov A., (2023), "Compression of GNSS Data with the Aim of Speeding up Communication to Autonomous Vehicles", Remote Sensing, MDPI, Vol. 15(8), paper no. 2165, available online at: https://www.mdpi.com/2072-4292/15/8/2165 , the author suggests in Figure 1 to interpolate the skipped data by an average from previous and subsequent information. I would encourage the authors to cite this paper in the related work section and explain what the advantages and disadvantages of this method are (compared to their methods).
Thank you for the suggestion. We believe directly including reference to this paper might dilute the discussion as currently it is not clear how one could use our accelerator circuits for (especially lossless) compression. Nonetheless, we will keep this paper in mind when publishing our compression coder/decoder accelerator work.

8. The equations and the results seem to be detached. Please explain how you have designed the experiments based on the theoretical background.
Thank you for your suggestion, we have made numerous updates throughput the paper, which should help in comprehending the proposed approach.

9. The format of references should be consistent.
Thank you! We have added the missing doi/url for the 9th reference: Chiew, Lin, Soon "A Novel Embedded Interpolation Algorithm with Negative Squared Distance for Real-Time Endomicroscopy"

Round 2

Reviewer 1 Report

Comments and Suggestions for Authors

I think that the article has been improved enough to be published.

Author Response

Thank you for the thorough review of the paper, we appreciate your help in improving the article.

Reviewer 2 Report

Comments and Suggestions for Authors

The authors submitted a sloppy version. There is no marking in red or any other color for the changes made. Their response is arguing with the comments instead of trying to improve the paper.

The authors refused to add an explanation of what each arrow means in Figures 4 and 5, refused to explain in the paper itself what "a" is, refused to add a reference and so on.

If the authors want to publish without peer review, they can simply put the paper on their website.

Still, I would recommend giving them another chance to improve the paper.

Author Response

Dear Reviewer,

We are sorry that the updated manuscript failed to uphold your well-argumented suggestions. While reviewing the updated version, it became apparent that there might have been some misunderstanding. Please refer to the updated version of the manuscript. We have sincerely tried to improve it. Here are some further clarifications.

- "The authors submitted a sloppy version. There is no marking in red or any other colour for the changes made. Their response is arguing with the comments instead of trying to improve the paper."
We have sincerely tried to improve the manuscript. We updated figures for readability, added clarifying notes on the results and revised the final latency and bandwidth considerations. As for the red markings, our understanding, following our experience in many other submissions, is that we are finalizing the paper, i.e., the paper will be published as is after the review.

- "The authors refused to add an explanation of what each arrow means in Figures 4 and 5, refused to explain in the paper itself what "a" is, refused to add a reference and so on."
There must have been some misunderstanding as following your initial feedback, we have added explicit clarification of the above-mentioned arrows in the Figure captions: "Arrows denote the direction of the data flow." Further, "a" is explicitly explained in the paper (also in the initially submitted version). See the first sentence of 4.1 section (3.1 before in the initial submission): "Typically, the corresponding memory address for the given coordinates adheres to the equation ...", which is followed by the equation defining "a" and explicit "a denotes address", hence "a" is the corresponding memory address to retrieve image data given specific row and column indices.

- We have also identified our technologies' potential application for the compression of real-time sensitive use cases, e.g. autonomous driving (Rakhmanov and Wiseman).

Hopefully, we have managed to address the paper's shortcomings in the updated version,
- Authors

Round 3

Reviewer 2 Report

Comments and Suggestions for Authors

The authors made a decent effort and the paper is certainly publishable so I would recommend accepting the paper.